# NF-κB in Cell Deaths, Therapeutic Resistance and Nanotherapy of Tumors: Recent Advances

**DOI:** 10.3390/ph16060783

**Published:** 2023-05-24

**Authors:** Xuesong Wu, Liang Sun, Fangying Xu

**Affiliations:** 1Department of Pathology, The First Affiliated Hospital, Zhejiang University School of Medicine, Hangzhou 310003, China; xuesong_wu@zju.edu.cn; 2Key Laboratory of Disease Proteomics of Zhejiang Province, Zhejiang University, Hangzhou 310058, China; 12218200@zju.edu.cn; 3Department of Pathology and Pathophysiology, and Department of Hepatobiliary and Pancreatic Surgery of the Second Affiliated Hospital, Zhejiang University School of Medicine, Hangzhou 310005, China

**Keywords:** NF-κB, apoptosis, necroptosis, pyroptosis, ferroptosis, therapy resistance, nanotherapy

## Abstract

The transcription factor nuclear factor-κB (NF-κB) plays a complicated role in multiple tumors. Mounting evidence demonstrates that NF-κB activation supports tumorigenesis and development by enhancing cell proliferation, invasion, and metastasis, preventing cell death, facilitating angiogenesis, regulating tumor immune microenvironment and metabolism, and inducing therapeutic resistance. Notably, NF-κB functions as a double-edged sword exerting positive or negative influences on cancers. In this review, we summarize and discuss recent research on the regulation of NF-κB in cancer cell deaths, therapy resistance, and NF-κB-based nano delivery systems.

## 1. Introduction

Therapeutic resistance to chemotherapy [1], radiotherapy [2], immunotherapy [3], and endocrine therapy [4], is currently a main obstacle in tumor therapy. Previously, various drugs exert their cancer-inhibiting effects by inducing cell death in various cancer types, and much of the therapeutic resistance is precisely due to the inhibition of apoptosis [5]. In the past decades, an increasing number of studies have focused on the induction of several types of programmed cell deaths (PCDs) [6,7]. A deep understanding of the characteristic and regulation of PCDs has promoted research on novel anti-cancer therapies targeting PCDs [8,9,10]. PCDs such as ferroptosis [11], necroptosis [12], and pyroptosis [13] are immunogenic cell deaths, which not only inhibit cell proliferation but also trigger immune responses and subsequently alter the immune microenvironment [14]. An in-depth comprehension of the pivotal signaling pathways associated with both PCDs and immune responses could contribute to attractive clinical therapeutic strategies.

Nuclear factor-κB (NF-κB), as a transcription factor that mediates many biological processes, is involved in multiple forms of immunogenic cell deaths. NF-κB remains at the cytoplasm in the resting condition and enters the cell nucleus when it is activated by various stimuli, thereby regulating diverse physiological and pathological processes including cell proliferation, angiogenesis, tumor progression, cell deaths, therapeutic resistance, inflammation, and immune responses [15]. It is well established that NF-κB is a significant endogenous proinflammatory factor. In recent decades, numerous high-quality literatures have summarized the molecular mechanism and function of NF-κB in regulating inflammation during tumorigenesis and progression [15,16,17,18,19,20,21,22].

Nano delivery systems (NDS) exhibit great promise for application in tumor treatment because they could accurately target tumor tissues, prolong the half-life of drugs, inhibit therapy resistance, reduce chemotherapy agents toxicity and side effects, and achieve drug combination regimens [23,24,25]. Nanotherapy based on NF-κB inhibition has attracted wide attention in recent years [26]. This review also summarizes the anti-tumor effects and mechanisms of the nanoparticulate systems inhibiting NF-κB. Here, we reviewed the effect and regulatory mechanism of NF-κB in tumor cell deaths, therapeutic resistance, and nanotherapy.

We searched the high-impact literature and examined thousands of titles and abstracts and hundreds of full texts through the PubMed and Web of Science databases. Specifically, we conducted data collection using the MeSH terms “NF-κB” and “tumors/cancers” combined with text words such as “cell deaths”, “treatment resistance”, or “nanotherapy”.

## 2. NF-κB Family Members and NF-κB Signaling Pathways

NF-κB is composed of five protein subunits including p50, p52, p65, RelB, and c-Rel [18]. These proteins permit DNA combination, dimerization, and nuclear translocation. Notably, p65, RelB, and c-Rel subunits display a transcriptional activation for DNA stimulation, while the p50 and p52 subunits merely serve as helpers for the DNA-binding effect of NF-κB [27,28,29]. P50 and p52 are stemmed from predecessors of p105 and p100, respectively [20]. Theoretically, NF-κB members are capable to form any combination of homodimers or heterodimers. However, not every hypothetically dimer could occur. The most common form of NF-κB is the p50/p65 heterodimer [15]. The NF-κB signaling pathway is tightly regulated under physiological conditions. NF-κB has mild nuclear activity under resting conditions. NF-κB could be stimulated by plenty of extracellular factors, which exhibit two different signal transduction cascades comprising the canonical and the non-canonical pathways [30,31].

In the canonical pathway, NF-κB activity remains at a low level through firmly fastening to the IκB that could mask the nuclear localization sequence of NF-κB [32]. The canonical NF-κB signaling is activated by TNF-α, TLRLs such as lipopolysaccharide (LPS), and IL-1. The main process of NF-κB activation is the proteasomal degradation of IκB through its phosphorylation by an inhibitor of κB kinase (IKK) complex containing two catalytic elements (IKKα, and IKKβ) and a regulatory element (IKKγ) [33]. Upon activation, the NF-κB heterodimer p50/p65 or p50/c-Rel is translocated to the cell nucleus, where it combines with a special DNA sequence and promotes gene transcription [34]. In the non-canonical pathway, NF-κB signaling is initiated through stimuli including BAFF, CD40L, RANKL, and LTβ. These extracellular factors stimulate the non-canonical pathway that is separate from IKKβ and IKKγ (also named NEMO) and IκB. Alternatively, NIK-induced phosphorylation of an IKKα homodimer mediates the procedure of p100 to p52, leading to the translocation of heterodimer p52/RelB into the cell nucleus and activating target gene expression [35,36,37,38]. Figure 1 displays the activation of the canonical and the non-canonical NF-κB pathways.

## 3. NF-κB Function in Tumor Cell Death

### 3.1. NF-κB Interacts with Apoptosis and Necroptosis

Apoptosis refers to the active and orderly cell death under physiological or pathological conditions in order to maintain the homeostasis of the internal environment. The morphological characteristics of apoptosis include cell shrinkage, nuclear condensation, nuclear DNA fragmentation, and the formation of apoptotic bodies. Notably, the cell membrane structure is intact during apoptosis [5]. Necroptosis is a highly regulated programmed necrosis and it has been extensively studied in recent years [12]. Cells undergoing necroptosis exhibit swelling of organelles, disassembly of the cytoplasm and nucleus, and rupture of the cellular membrane [39] promoting the release of cell contents and inducing inflammatory response [40], which is distinguished from apoptosis.

It is generally believed that NF-κB inhibits cancer cell apoptosis and most types of tumors are resistant to TNF-α/TNFR-mediated apoptosis because of the preceding activation of NF-κB [41]. NF-κB could upregulate anti-apoptotic molecules, such as Bcl-2, Bcl-XL, and FLIP [42], and enhance the transcription of anti-oxidant enzymes such as MnSOD and FHC to eliminate the level of ROS in tumor cells [43], which eventually exert the function of apoptosis inhibition. Consistent with this, depression of NF-κB could convert the function of TNF-α signaling from survival to death in ovarian cancer cells [44] and promote TNF-α-induced CCA, apoptosis, and necroptosis in hepatocellular carcinoma (HCC) cells [45]. Moreover, NF-κB suppression could potentiate apoptosis and further inhibit the development of non-small cell lung cancer (NSCLC) cells [46]. These results suggest that tumor cells lacking NF-κB signaling are more vulnerable to apoptosis and, therefore, have a lower malignant potential [47].

Understanding the molecular mechanisms of apoptosis and necroptosis is important to clarify how NF-κB interacts with them. TNF-α could provoke several cell death processes, including apoptosis and necroptosis. First, the combination of TNF-α with TNFR promotes the combination of RIPK1, TRADD, cIAP, and TRAF and, thus, the constitution of complex I. Second, complex II comprised of RIPK1, TRADD, caspase 8, and FADD is formed when RIPK1 is deubiquitinated by CYLD. Third, caspase 8 of complex II inactivates RIPK1 by proteolytic cleavage, which finally induces the apoptosis pathway [44]. When caspase 8 is blocked, RIPK1 is stabilized and exerts the function of phosphorylated to activate RIPK3 and form a vital protein complex named necrosome with it. Subsequently, RIPK3 phosphorylates its substrate pro-necrotic enzyme MLKL, promoting its oligomerization and translocation to the cytomembrane and promoting membrane disruption to execute necroptosis [12].

RIPK1 [42] and caspase 8 [48] are vital molecular switches in regulating the equilibrium between cell survival and cell death, and both are implicated in the relation between NF-κB and apoptosis as well as NF-κB and necroptosis. Recent research has found that RIPK1 recruits to the IKK complex and activates NF-κB, leading to NF-κB binding to the promoters of CXCR4 and uPA, then facilitating breast cancer metastasis [49]. The ubiquitination of RIPK1 by cIAP in complex I activate TAK1, which phosphorylates IKK and subsequently facilitates NF-κB activity and cell survival [50,51]. Moreover, suppression of RIPK1 could impair NF-κB and potentiate HCC progression [52]. These results demonstrate that RIPK1, an important molecule involved in apoptosis, could induce NF-κB activation and promote tumor development.

Since NF-κB could be augmented by RIPK1 [53], caspase 8 inhibitors lead to NF-κB activation and resist apoptosis by sufficiently blocking the cleavage of RIPK1 and stabilizing the expression of RIPK1 [54]. Additionally, the inhibition of caspase 8 also activates NF-κB by stimulating the RIPK1/RIPK3/MLKL signaling pathway. Specifically, the axis activates the MAPK pathway and promotes IκB degradation and the sustained nuclear entry of NF-κB, thereby increasing the release of inflammatory cytokine during necroptosis, such as CXCL1, CXCL8, CCL20, and CSF1 [55]. Moreover, RIPK3-phosphorylated TRIM28 could upregulate NF-κB in tumor cells, leading to elevated immunostimulatory cytokine such as GM-CSF expression, thereby contributing to robust cytotoxic anti-tumor immunity [56]. These findings indicate that NF-κB might be prevented by caspase 8, a co-regulator of apoptosis and necroptosis.

Suppression of caspase 8, which is a central molecule associated with cell death [48], could facilitate the necroptosis of ovarian cancer cells [44]. Moreover, pan-caspase inhibitors could facilitate 5-FU-induced necroptosis in colorectal cancer (CRC) cells [57]. NF-κB is considered an apoptosis inhibitor and various cancers are resistant to apoptosis, owing to the previous activation of NF-κB [41]. Inducing necroptosis is expected to be an effective strategy to prevent tumor progression and improve chemotherapy sensitivity when tumor cell apoptosis is inhibited by NF-κB. Therefore, the induction of necroptosis has been recommended as an approach to defeat apoptosis resistance.

### 3.2. NF-κB and Pyroptosis

Pyroptosis is described as GSDM-mediated and inflammatory PCD, accompanied by plasma membrane perforation and cell contents release, which then promotes an immune response and inflammation [58]. After cleavage by caspases or granzymes, the GSDMs cause the cells to swell, rupture and undergo pyroptosis [14,59,60,61,62]. Pyroptosis could play a double role in potentiating and suppressing cancer cell growth [63]. Our previous study found that nuclear GSDMD promotes CRC invasion and metastasis, while patients with elevated cytoplasmic GSDMD expression have a reduced danger of distant metastasis and improved clinical outcome, indicating that the role of GSDMD during tumor progression depends on its subcellular locations [64].

Recently, several studies have found that intracellular molecules, or extracellular compounds that inhibit NF-κB signaling, could promote tumor cell pyroptosis. For instance, DRD2 restricts the NF-κB signaling pathway as well as induces pyroptosis, and regulates the tumor microenvironment to prevent breast cancer [65]. Similarly, Tanshinone II could prevent NF-κB and significantly upregulate the caspase 3/GSDMD pathway, thereby promoting pyroptosis and preventing cervical cancer progression [66]. Piperlongumine, a bioactive alkaloid extracted from plants, prevents various biological functions containing anti-cancer and anti-inflammatory. It is found that the piperlongumine analog L50377 triggers pyroptosis through ROS-mediated NF-κB inhibition in NSCLC [67]. These studies suggest that NF-κB may impede pyroptosis and promote tumor development.

On the contrary, it has also been demonstrated that NF-κB could promote pyroptosis and inhibit tumor progression. For example, Polyphyllin VI, a kind of traditional Chinese medicine, stimulates pyroptosis by activating the NF-κB/NLRP3/caspase 1/GSDMD pathway in NSCLC [68]. Moreover, NF-κB facilitates Bax activation and cytC release and further induces the activity of caspase 3 and degradation of GSDME to promote cell pyroptosis and exert anti-cancer effects in HCC, breast cancer, and CRC under the stimulation of the metformin-enhanced AMPK/SIRT1 pathway [69]. The initiation of pyroptosis via metformin/NF-κB is regarded as an innovative therapeutic choice to prevent diverse tumors. In addition, NF-κB facilitates GSDMD transcription through binding to the GSDMD promoter area and elevates GSDMD expression to induce the pyroptosis of adipocytes under LPS stimulation [70].

These studies suggest that NF-κB might play a dual pro-tumor and anti-tumor role by inhibiting or promoting GSDM-mediated pyroptosis, respectively. Interestingly, GSDM regulates NF-κB expression in turn and plays a key role in steatohepatitis [71]. However, no relevant studies focus on whether NF-κB could be regulated by GSDMs in tumors at present.

### 3.3. NF-κB Regulates Ferroptosis

Ferroptosis is characterized by iron overload and lipid peroxidation [8,10]. GPX4, an essential GPX that decreases lipid peroxidation, serves as a crucial inhibitor of ferroptosis. The ferroptosis activator RSL3 induces ferroptosis in several cancers through decreasing GPX4 expression [8]. Cystine/glutamate antiporter system xc−, whose primary component is SLC7A11, functions as a cystine/glutamate antiporter to formulate GSH. GSH is converted into GSSG under the catalysis of GPX4, neutralizing the oxidative substances in the plasma membrane, and thus inhibiting ferroptosis.

NF-κB plays a dual anti-ferroptosis and ferroptosis-promoting role, depending on the tumor types or the specific stimulus. Most of these studies focus on the negative function of NF-κB on ferroptosis. A recent report demonstrated that DMF inhibits NF-κB signaling and efficiently induces lipid peroxidation and ferroptosis in diffuse large B cell lymphoma (DLBCL) cells and exerts a broad anti-lymphoma effect [72], indicating that NF-κB may associate with anti-ferroptosis behavior. NKAP protects glioblastoma (GBM) cells from ferroptosis by upregulating SLC7A11 and promoting cell resistance to ferroptosis inducers [73]. In addition, NF-κB could prevent ferroptosis through directly increasing ferroptosis-negative regulator GPX4 or enhancing iron-sequestering molecular LCN2. Researchers identify GPX4 as a direct target of NF-κB using RNA-sequencing and bioinformatic analyses [74]. Moreover, NF-κB is activated by the prevention of the LIFR/SHP1 axis could enhance the expression of LCN2, thereby exhausting iron and causing resistance to ferroptosis to promote liver tumorigenesis [75]. These studies show that NF-κB leads to ferroptosis suppression and tumor progression by preventing lipid peroxidation as well as reducing cellular iron. Notably, an LCN2-neutralizing antibody increases the ferroptosis-potentiating and anti-tumor effects of sorafenib (SOR), a chemotherapeutic agent that could inhibit the system xc− and induce ferroptosis, in HCC patient-derived xenograft tumors [75]. NF-κB could be inhibited by iron chelator DFX. Intriguingly, even though ferroptosis is inhibited by DFX, the SOR and DFX union show accumulated anti-cancer effects for HCC via apoptosis and NF-κB signal regulation [76].

NF-κB may also promote tumor cell ferroptosis and reduce tumor malignancy. Several studies have shown that specific bioactive molecules or exogenous compounds could facilitate NF-κB signaling as well as promote ferroptosis to eliminate tumor aggravation. For example, as an inflammation mediator, HMGB1 not only enhances the LPS-induced NF-κB to intensify the inflammation in colon cancer cells but also downregulates GPX4 activity and thereafter promotes ferroptosis [77]. The AMPK/NF-κB pathway is modulated by a plant-derived triterpenoid lupeol that is found to decrease GPX4 and GSH levels, trigger ferroptosis, and suppress nasopharyngeal carcinoma (NPC) [78]. NF-κB is induced by RSL3 in the GBM cells, and NF-κB suppression could mitigate RSL3-induced ferroptosis [79], suggesting that NF-κB is critical for RSL3-induced ferroptosis and subsequent GBM suppression. NF-κB also facilitates the ferroptosis of GBM by downregulating SLC7A11 [79].

Ferroptosis is a recently defined PCD process in 2012, few researchers have concentrated on the regulation of NF-κB on ferroptosis, and further research is needed to deeply understand the relationship between NF-κB and ferroptosis. Figure 2 illustrates the crosstalk between NF-κB and cell deaths including apoptosis, necroptosis, pyroptosis, and ferroptosis.

## 4. Association of NF-κB with Therapeutic Resistance

### 4.1. NF-κB Induces Chemotherapy Resistance

Chemoresistance refers to the insensitivity of tumors to chemotherapy drugs and the unsatisfactory effect of treatment. Chemoresistance is a major challenge resulting in cancer recurrence and an unfavorable prognosis [80]. Recent findings demonstrate that NF-κB regulates chemotherapy resistance in various tumors, including gastric cancer [81,82], CRC [83], NSCLC [84,85,86], pancreatic ductal adenocarcinoma (PDAC) [87], breast cancer [88,89,90,91,92], prostate cancer [93], GBM [94,95], and hematological tumors [96,97,98,99].

Microorganism is involved in the onset and progression of diverse human diseases, including cancer. Microbiota mainly exists in the gastrointestinal tract, and its products, such as LPS, can contribute to gastrointestinal tumorigenesis by regulating NF-κB signaling [100]. Helicobacter pylori (H. pylori) is regarded as one of the most common pathogenic factors of gastric cancer and its infection contributes to microbial dysbiosis that may participate in gastric carcinogenesis and development [101]. H. pylori could upregulate RASAL2 transcriptional expression through NF-κB activation. Overexpression of RASAL2 is common in gastric cancer and could contribute to gastric tumorigenesis, poor prognosis, and the chemoresistance of platinum and fluorouracil-based chemotherapy by activating the AKT/β-catenin pathway that could be suppressed by PP2A [82]. Moreover, NF-κB inhibition may reverse cisplatin resistance in gastric cancer cells [81] and restore chemosensitivity including oxaliplatin and vincristine in CRC cells by preventing NRF2/MRP2 axis [83] that might enhance multidrug resistance in various tumors. These studies suggest that NF-κB could induce chemoresistance in gastrointestinal tumors, including gastric cancer and CRC.

Previous research has indicated that NF-κB promotes EGFR-TKI’s resistance to NSCLC [86] and NF-κB inhibition could facilitate EGFR-TKI’s sensitivity [84]. Overcoming acquired resistance to EGFR-TKIs, such as osimertinib and erlotinib, would be a major breakthrough in the treatment of NSCLC. Osimertinib-induced TGF-β2 activates NF-κB in NSCLC cells and upregulates the SMAD2/EMT axis, which may further promote osimertinib resistance [85]. It is well known that miRNAs could participate in many tumor-related biological processes, including metastasis and chemoresistance, through binding to the target gene and inhibiting special gene expression. NF-κB eliminates the miR-590 expression and upregulates EHD1, thus increasing stem cell-like properties and erlotinib resistance in NSCLC [84]. MiR-135b could suppress deubiquitinase CYLD, a negative regulator of NF-κB, leading to NF-κB activation, NSCLC progression, and an unfavorable prognosis [102]. NF-κB directly inhibits the transcriptional expression of miR-488 which could target and prevent the expression of ERBB2, leading to the growth and malignancy of pancreatic cancer cells [103]. Intriguingly, miR-146a-5p could downregulate the TRAF6/NF-κB axis and drive gemcitabine chemoresistance in PDAC [87]. Although miRNA is generally regarded as a novel therapeutic strategy, since it could specifically prevent tumor-related genes [104], the relationship between NF-κB and miRNA is complicated. MiRNA interacts with NF-κB and eventually exerts the function of tumor promotion or prevention.

Moreover, NF-κB is related to the chemoresistance of breast cancer. Breast cancer with high NF-κB activity is more likely to develop chemotherapy resistance. Additionally, residual treatment-refractory breast cancer tissues show high levels of NF-κB [90]. MiR-1910-3p targets and inhibits MTMR3 and activates the NF-κB signaling, thereby potentiating breast cancer proliferation and metastasis [105]. MiR-132 and miR-212 significantly inhibit the expression of PTEN, which could promote the AKT/NF-κB pathway and further reduce the sensitivity of breast cancer cells to doxorubicin (DOX) [88]. Targeting the NF-κB/JNK axis might eliminate remaining cancer cells and promote treatment efficiency by inducing the apoptosis of breast cancer cells [90]. ATM and DNA-PKcs, as activators of NF-κB, are critical to the response of cytotoxic chemotherapies [91]. Etoposide and cisplatin induce DNA-PKcs and ATM-mediated activation of NF-κB, enhancing APOBEC3B expression, which drives tumor progression and therapy resistance of breast cancer [92]. In addition, post-translational modifications also play a vital role in regulating NF-κB signaling activity [106]. The ubiquitination and degradation of NF-κB by FBXW2 could decrease SOX2 expression and suppress breast cancer stemness, tumorigenesis, and paclitaxel resistance [89].

Gut microbiota participates in the development and chemotherapy resistance of several tumors [107,108,109]. LPS could promote the progression and docetaxel resistance of prostate cancer by activating the NF-κB/IL6/STAT3 signaling pathway [93].

In addition, aberrant NF-κB activation is prevalent in GBM and could affect both tumor development and chemoresistance. The routine therapy options for patients with GBM contain surgical operation, radiotherapy, and chemotherapy with temozolomide (TMZ) [110], however, long-term use of TMZ may lead to chemoresistance [79]. ADAR3 upregulates NF-κB expression and eliminates GBM cell sensitivity to TMZ [95]. Moreover, chemotherapy promotes ROS generation that could facilitate tumor cell survival through activating NF-κB, thereafter upregulating the anti-apoptotic gene Bcl-XL and enhancing GBM survival and chemotherapy resistance [94].

Notably, NF-κB also participates in chemotherapy resistance in hematologic tumors, such as multiple myeloma (MM) and acute lymphoblastic leukemia (ALL). MM is one of the most frequently occurring hematologic malignant tumors and proteasome inhibitor bortezomib is the first-line therapy for newly diagnosed MM [111]. MM stem cells are considered to be the primary reason for chemoresistance and recurrence in patients with MM. NF-κB inhibition could suppress bortezomib resistance and tumor stemness in MM [99]. Moreover, HAPLN1 and MMP2 from bone marrow stromal cells activate NF-κB and induce resistance to bortezomib treatment in MM [97]. ALL remains the main reason for death among all pediatric tumors [112]. The BM niche is a complicated microenvironment and may participate in the chemoresistance of ALL [113,114]. Chemotherapeutic drugs, including Ara-C, DNR, and 6-MP, trigger NF-κB activation by stimulating the ATM/TRAF6 signaling pathway, which directly upregulates bone marrow niche-protecting cytokines GDF15, CCL3, and CCL4, thereby leading to drug resistance of ALL [98]. Consistently, suppression of the ATM-dependent NF-κB signaling could improve the sensitivity of ALL to chemotherapy. Mantle cell lymphoma (MCL) is a refractory aggressive lymphoma with an adverse clinical outcome. MiR-223-3p could directly inhibit conserved helix-loop-helix ubiquitous kinase (CHUK) and further decrease the expression of NF-κB, thereby reversing Bruton’s tyrosine kinase (BTK) inhibitor ibrutinib resistance in MCL [96].

Furthermore, IKKα participates in NF-κB-mediated chemotherapy resistance and could function as a novel target for enhancing chemotherapy effectiveness [115]. It is well known that IκB is downgraded in the ubiquitin-proteasome system, liberating and enabling NF-κB nuclear translocation and activation. Therefore, suppression of the UPS may block NF-κB and, therefore, enhance the chemotherapy sensitivity of malignant tumors [116].

In a word, chemoresistance is a primary difficulty in cancer treatment, and NF-κB inhibition is expected to promote tumor treatment efficacy, improve clinical outcomes, and extend the survival time of patients.

### 4.2. NF-κB Promotes Radiotherapy Resistance

Radiotherapy, also named IR, has the advantage of relatively fewer systemic side effects compared to chemotherapy. Radiotherapy is one of the primary treatments for squamous cell carcinoma and GBM. NF-κB plays a vital role in the radiotherapy resistance of cancers and the suppression of NF-κB promotes sensitivity to radiotherapy. Radiotherapy resistance is the main reason that causes therapy failure and the recurrence of NPC. A recent study has shown that AKR1B10-induced free fatty acid synthesis potentiates the TLR4/NF-κB axis which then regulates CCA and DDR, leading to radiation insensitivity of NPC [117]. NF-κB is also a critical driver of radioresistance in human head-and-neck squamous cell carcinoma [118]. IR-induced cell death is one of the most common mechanisms for the radiotherapy of several tumors. However, IR also could activate NF-κB that in turn inhibits IR-mediated apoptosis of esophageal cancer cells [119]. PELI1, a NIK E3 ubiquitin ligase, mediates the degradation of NIK and then suppresses the activation of the IR-stimulated NF-κB, causing the downregulation of Bcl-XL, thereby promoting the apoptosis of esophageal cancer cells induced by radiotherapy [119], indicating that activation of the NIK/NF-κB pathway could promote radioresistance by inhibiting apoptosis.

Radiotherapy is still the standard treatment approach in patients with GBM [120]. Nevertheless, continual radiation activates NF-κB and then upregulates the expression of YY1 which could directly suppress miR-103a transcription, resulting in activating the DDR and promoting stemness and the expansion of tumor cells, thereby driving radioresistance and the recurrence of GBM [121].GBM is extremely heterogeneous and contains three subtypes, namely proneural (PN), mesenchymal (MES), and classical [122]. PMT is a critical phenotypic transformation in GBM and is tightly related to therapy resistance, tumor recurrence, and poor clinical outcome [110]. NF-κB is proven to be the main molecule that regulates PMT. Recent research has found that NF-κB could be upregulated by FOSL1 which plays a crucial role in cell proliferation, differentiation, and survival, and eventually promotes PMT and radioresistance [122]. Moreover, the protein complex ARPC1B-TRIM21-IFI16 activates NF-κB and promotes PMT and radiotherapy resistance [123]. These results demonstrate that NF-κB could induce PMT in GBM and thereby reduce radiosensitivity.

In addition, NF-κB also promotes radiotherapy tolerance in CRC by inhibiting tumor cell apoptosis. ALDH1L2 facilitates a redox protein TXN degradation and activates NF-κB, thereby upregulating CAT and SOD2, and eliminating ROS to protect cells from ROS-mediated apoptosis, thereby inducing radioresistance in CRC cells [124].

These studies indicate that NF-κB plays an essential role in the radiotherapy resistance of tumors including NPC, HNSCC, esophageal cancer, GBM, and CRC.

### 4.3. NF-κB and Endocrine Therapy Resistance

In addition to chemotherapy resistance and radiotherapy resistance, NF-κB is also involved in endocrine therapy resistance. Endometrial cancer, breast cancer, and prostate cancer are common sex hormone-related cancers that are associated with estrogen, progesterone, and androgen respectively, and could be treated with endocrine therapy. NF-κB is a significant mechanism for therapy resistance of the three types of cancers [125,126].

Studies have shown that nearly 75% of breast cancer express ER. TAM could antagonize estrogen by competitively binding to ER, leading to the suppression of ER-positive breast cancer [127]. NF-κB is a driver of TAM resistance [128,129,130]. For example, PKC-ε and PKD3 are interacted with TRIM47 and enhance NF-κB signaling, leading to breast cancer proliferation and TAM resistance [130]. Moreover, HMGB1 leads to TAM endocrine therapy insensitivity by combining with TLR-4 and inducing NF-κB activation [128]. CDK4/6 kinase suppressors, such as palbociclib, ribociclib, and abemaciclib may inhibit HMGB1 and decrease the TLR-4/NF-κB axis, and reverse TAM resistance [128]. Resistance to endocrine therapy is a primary obstacle to the treatment of breast cancer. NF-κB is engaged in TAM tolerance and inhibition of NF-κB is expected to improve the sensitivity of TAM treatment. Moreover, NF-κB upregulates the anti-apoptotic factor of BCL-2 to exert the effect of endocrine resistance in ER-positive breast cancer [131]. The NF-κB pathway could be activated in endocrine therapy-resistant breast cancer [132] and may cause more invasive disease and eventual relapse [133]. Intriguingly, ER is an inhibitor of NF-κB, and prevention of ER leads to the reactivation of NF-κB [134], however, whether the inhibitory effect of ER on NF-κB is related to the resistance of NF-κB to endocrine therapy is still inconclusive.

Except for estrogen antagonists TAM, endocrine treatment options for ER-positive breast cancer also recommend aromatase inhibitor anastrozole and ovarian ablation or suppression [135]. Aromatase depressors could improve clinical outcomes compared with TAM in postmenopausal women with breast cancer [136]. However, aromatase inhibitors are proven to be associated with side effects that reduce patients’ quality of life [137]. At present, there is no direct evidence that NF-κB engages in aromatase inhibitor resistance in breast cancer.

AR plays an essential role in the proliferation and development of prostate cancer. Consequently, ADT is used to treat prostate cancer patients. NF-κB is associated with castration-resistant prostate cancer and castration could activate NF-κB [138]. HMGB1 is a significant factor in the progression and invasion of prostate cancer. HMGB1 not only promotes TAM resistance in breast cancer but also participates in castration resistance in prostate cancer. HMGB1, combined with TNFR1, facilitates cancer progression and castration resistance by inducing NF-κB activation in prostate cancer [139]. Furthermore, NF-κB-induced resistance to ADT may be mediated by the excessive activation of AR and production of AR variant-7 in prostate cancer cells. Therefore, reversing resistance to ADT could be reached by utilizing an oral NF-κB inhibitor DMAPT [140].

Another hormone-related tumor is endometrial cancer. Conservative progesterone-based treatment is an efficient therapy for relapsed or refractory endometrial cancer. Activation of NF-κB also associates with progesterone resistance in endometrial cancer. The NF-κB pathway is activated by SREBP, a transcription factor that mediates the anabolism of cholesterol and lipids, and then inhibits apoptosis, and boosts the proliferation and resistance to progesterone of endometrial cancer cells. Therefore, as an inhibitor of SREBP1, fatostatin can improve the sensitivity of endometrial cancer to progesterone and reverse progesterone resistance by suppressing SREBP1/NF-κB signaling [141].

### 4.4. NF-κB Functions in Immunotherapy

Evasion of immune surveillance is a hallmark of cancers [142]. While antigens derived from cancer cells are possibly identifiable by the immune system, cancer cells could evade immune attacks through different mechanisms, including abnormal activation of immune checkpoints that could terminate immune responses [143]. PD-1 and PD-L1 are the two main immune checkpoint proteins. PD-L1 expressed in tumor cells combines with its receptor, PD-1 expressed in T-cells, thereby inducing T-cell apoptosis, preventing cytotoxic T-cell efficacy, and triggering tumor immune evasion. The PD-1/PD-L1 blockade therapies (immune checkpoint block, ICB) have been approved for the therapy of diverse tumor types. Although anti-PD-1 and anti-PD-L1 antibodies could improve patient disease-free survival, the response among some patients is still unsatisfactory.

NF-κB is vital in modulating both innate and acquired immune responses. NF-κB is a significant regulator of the immune evasion of tumor cells [144]. NF-κB upregulates the transcriptional expression of PD-L1 [144,145] in several cancer types such as pancreatic cancer [144,146], cervical cancer [147], prostate cancer [148], and CRC [149]. NF-κB and PD-L1 could be suppressed by CDK4/6-phosphorylated RB protein, which is a well-studied tumor suppressor, thus eliminating tumor development, and this seems to be a common regulatory mechanism occurring in different tumor types [150]. Therefore, CDK4/6 inhibition or RB suppression might induce the undesirable activation of the NF-κB/PD-L1 pathway and cause an immune evasion of tumor cells [150]. HDAC5 prevents NF-κB and decreases PD-L1 expression, inhibiting cancer immune evasion, and leading to a favorable clinical outcome for patients with PDAC. Consistently, HDAC5 silencing could improve ICB sensitivity by activating the NF-κB/PD-L1 axis in immunotherapy-resistant pancreatic tumors [144], demonstrating that the effectiveness of ICB depends on the high expression rate of PD-L1 of tumor cells and NF-κB activation could reverse ICB resistance by enhancing PD-L1 expression. Furthermore, NF-κB could induce the transition of macrophage in the tumor microenvironment from M2 to M1 phenotype and decrease the level of inflammatory factors (IL-10, TGF-β and PGE2) which could prevent cytotoxic T-cells, thereby upregulating the expression of PD-1 in T-cells and potentiating the sensitivity to anti-PD-1 therapy of HCC [151]. Response to immune checkpoint therapy is closely associated with PD-L1 expression in tumor cells and PD-1 expression in T-cells. The activation of NF-κB upregulates the expression of PD-L1 in tumor cells and PD-1 expression in T-cells, which is a double-edged sword. The NF-κB/PD-L1 (or PD-1) pathway could lead to immune evasion on the one hand and elevate immunotherapy sensitization and reverse immunotherapy resistance on the other hand.

However, activation of the NF-κB/PD-L1 pathway also leads to resistance to ICB therapy. For example, ligation of TLR/IL-1R activates IRAK4, the primary kinase that could trigger the innate immune response, and activates the TAK1/IKK/NF-κB pathway, ultimately upregulating PD-L1 and HAS2, and driving dysfunction and the exhaustion of T-cell anti-tumor immunity and resistance to ICB [146]. Therefore, targeting IRAK4 could prevent PD-L1 and activate T-cells, therefore highly enhancing immunotherapy in pancreatic cancer. In addition, MGP upregulates NF-κB and then activates PD-L1 expression to facilitate cytotoxic T-cell exhaustion, thereby promoting liver metastasis of CRC. The union of MGP knockdown and anti-PD-1 therapy can synergistically resist liver metastasis of CRC [149]. Furthermore, tumor size decrease is significantly obvious when the combination of anti-PD-1 with NF-κB inhibition compared with anti-PD-1 therapy alone [152], suggesting that the optimal function of PD-1 suppressor demands prevention of NF-κB activity. These results show that the NF-κB/PD-L1 axis promotes immunotherapy resistance and the combination of anti-PD-L1 (or PD-1) with NF-κB suppression significantly promotes tumor regression [153].

Conservative treatments for cancers include chemotherapy, radiotherapy, endocrine therapy, and immunotherapy. NF-κB can be activated by chemotherapy, radiotherapy, and endocrine therapy, contributing to treatment failure and resistance in various cancer types. Therefore, inhibition of NF-κB can relieve therapeutic tolerance and improve the treatment sensitivity of tumors. However, the role of NF-κB in immunotherapy is complicated and needs to be further elucidated. Figure 3 demonstrates the relationship between NF-κB and therapeutic resistance.

The bioactive molecules or exogenous compounds which promote or inhibit NF-κB signaling are listed in Table 1.

## 5. NF-κB-Based NDS

As mentioned above, NF-κB activation generally prevents apoptosis and leads to treatment resistance in diverse tumors. NDS developed rapidly over the past decades. NDS could increase the transfer efficacy and enhance the bioavailability of agents, reduce drug toxicity and side effects, and eventually improve patients’ prognosis [26]. Recent findings demonstrate that anti-NF-κB nanotherapy has the potential to inhibit the progression, metastasis, and chemotherapy resistance in various tumors such as CRC [154], NSCLC [155], pancreatic cancer [156], breast cancer [157,158], GBM [159], adult T-cell leukemia/lymphoma (ATLL) [160], and melanoma [161].

NF-κB could be activated by anti-cancer agents DOX and then upregulate anti-apoptotic genes, exert pro-survival function, and further reduce DOX sensitivity [162]. Both selenium and curcumin constituents could inhibit NF-κB activation [163]. Therefore, curcumin-loaded selenium nanoparticles (Se-Cur NPs) are developed and applied in combination with HA-based polyethylene glycol (PEG) and related NPs loaded with DOX (PSHA-DOXNPs). The NPs complex prevents the expression of NF-κB and induced CCA and apoptosis in CRC cells through Se-Cur NPs and enhances the efficacy of DOX at the same time [154].

Furthermore, researchers devised a hyaluronic acid (HA)-modified glycol chitosan (GC) nanoparticle that carried DOX and celecoxib (CXB), it is named HA-GC-DOX/CXB. The HA-decorated NPs effectively carried agents into NSCLC via CD44-mediated endocytosis. The HA-GC-DOX/CXB system obviously downregulates cyclooxygenase (COX), MMPs, and NF-κB, as well as increases the expression of caspase 3, thereby significantly suppressing NSCLC tumor inflammation, proliferation, invasiveness, and enhancing apoptosis [155]. In addition, the HA-modified poly lactic-co-glycolic acid (PLGA)-PEG nanoparticles (PLGA-PEG-HA NPs) could target and deliver thiotetrazole, a suppressor of PI3K, to CD44 overexpressing pancreatic cancer cells, thereby improving the efficacy of anti-cancer drugs and inducing tumor cell premature senescence through the inhibition of the PI3K/AKT/NF-κB signaling pathway in pancreatic cancer [156].

Chrysin is a flavone and has anti-cancer activity [164]. Hydrophobic poly (ε-caprolactone) PCL-PEG-NPs encapsulated chrysin could promote G protein-coupled ER (GPER) expression and lead to the inhibition of NF-κB/MMPs signaling, which could prevent tumor aggressiveness and metastasis in breast cancer [158]. Additionally, astaxanthin solid lipid NPs (AX-SLN) inhibit the AKT/NF-κB axis and then contribute to the inhibition of breast carcinogenesis and expansion [157].

NF-κB-based nano delivery systems also reduce the malignancy of brain tumors. Disulfiram is a well-known anti-alcohol drug and has anti-cancer activity by eliminating cancer stem cells (CSCs) and reversing chemoresistance. The cytotoxicity of disulfiram relies on copper and the disulfiram-copper complex could prevent NF-κB activation [165]. It is found that PLGA-encapsulated disulfiram (DS-PLGA) prevents NF-κB activation, eradicates GBM stem cells, blocks migration and invasion of GBM cells, and reverses chemoresistance [159].

Anti-NF-κB nanotherapy could effectively prevent hematological tumors such as ATLL [160]. Small interfering RNA (siRNA) could target mRNA for cancer treatment. A peptide-based NF-κB siRNA NP which prevents the expression of the NF-κB signaling pathway could be rapidly delivered to ATLL, thereby inhibiting ATLL growth and sensitizing advanced ATLL to etoposide [160]. Furthermore, siNF-κB could also be encapsulated in polymeric micelle NPs and carried to tumor tissues or metastatic sites via intravenous administration, thereby inhibiting melanoma migration/invasion and pulmonary metastasis [161].

Table 2 demonstrates a summary of anti-NF-κB nano delivery systems.

## 6. Conclusions and Future Outlook

NF-κB acts as a double-edged sword in cancer due to its multifaceted role such as regulating tumor metabolism, promoting angiogenesis, mediating inflammation, affecting cell death, and participating in treatment resistance. NF-κB is involved in the whole process of tumor evolution by facilitating inflammation in carcinogeneses and regulating the inflammatory state of the tumor microenvironment in tumor progression. Due to the limited space, we focused on the regulation of NF-κB in cell death and treatment resistance in various tumors. Cell death is a strictly regulated process that serves as a natural barrier that prevents the development and progression of cancer, and the resistance to cell death is often responsible for therapy tolerance and failure. Although NF-κB could induce therapy resistance by regulating cell deaths, no “one-fits-all” mechanism can be derived.

Notably, most studies support the role of NF-κB in tumor suppression. The endogenous active molecules, exogenous compounds, miRNAs, and natural plant extracts mentioned in this review could act as NF-κB regulators, affecting cell deaths, tumor development, and treatment resistance. As a novel drug delivery technique, nanoparticles could transport diverse bioactive/exogenous compounds that inhibit NF-κB activity and improve the efficacy of anti-cancer drugs. However, the clinical application of NDS is still limited, and large-scale clinical trials are needed to elucidate whether NDS-mediated NF-κB inhibition could improve the clinical benefit of cancer patients. Based on the intricate relationship between NF-κB and cancer, there is still a long way to go to achieve the benefits of precise treatment by targeting the NF-κB family.

## Figures and Tables

**Figure 1 pharmaceuticals-16-00783-f001:**
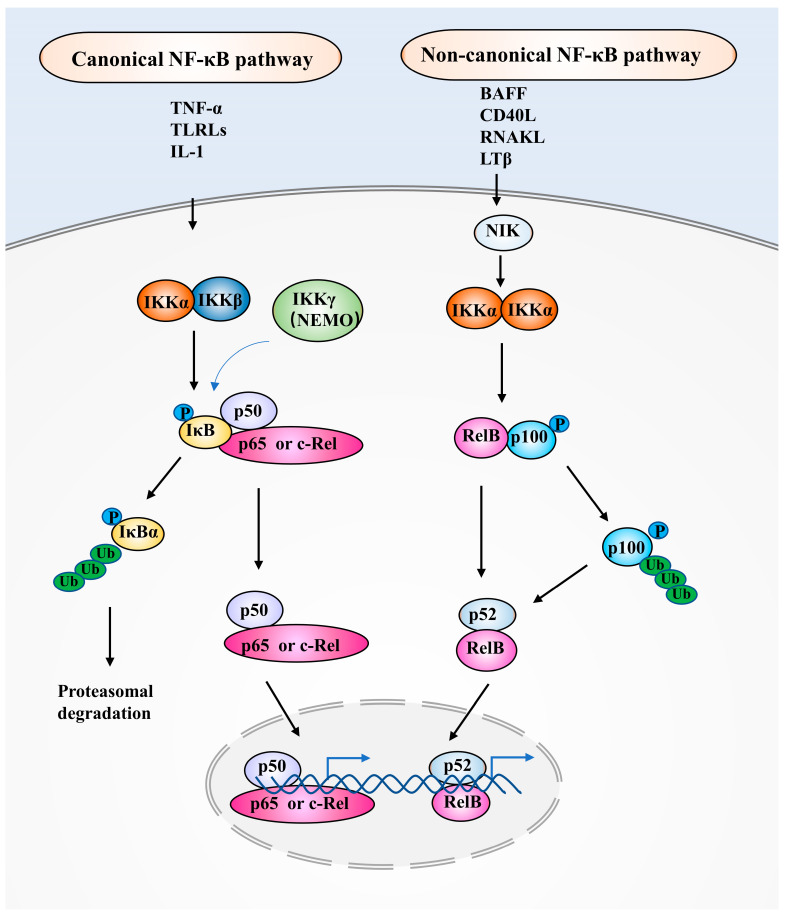
The two NF-κB signaling pathways. The activation process of the canonical and the non-canonical NF-κB pathways are shown on the left and right sides, respectively. RHD, Rel homology domain; IκB, inhibitor of κB; TNF-α, tumor necrosis factor-α; TLRLs, Toll-like receptor ligands; IL, Interleukin; IKK, IκB kinase; NEMO, NF-κB essential modifier; BAFF, B-cell-activating factor; CD40L, CD40 ligand; RANKL, receptor activator of NF-κB lig-and; LTβ, lymphotoxin β; NIK, NF-κB inducing kinase.

**Figure 2 pharmaceuticals-16-00783-f002:**
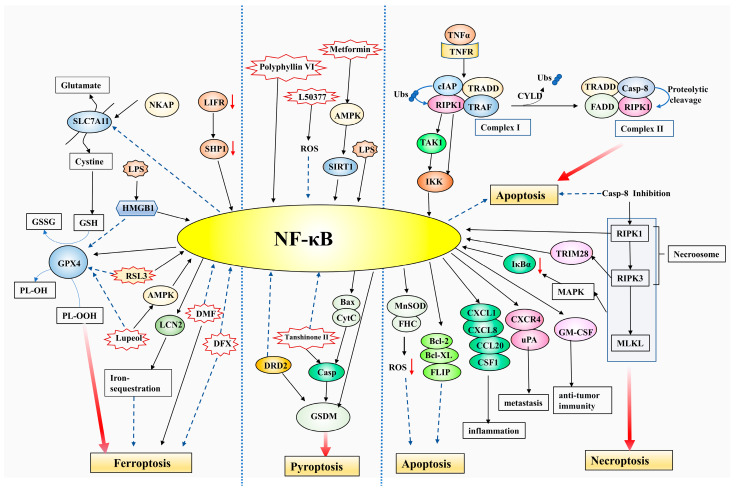
NF-κB functions in several types of PCD containing apoptosis, necroptosis, pyroptosis, and ferroptosis. Black solid arrows represent facilitation and blue dashed arrows represent inhibition. TNFR, TNF receptor; FLIP, FLICE-inhibitory protein; MnSOD, manganese superoxide dismutase; FHC, ferritin-heavy chain; RIPK1, receptor-interacting protein kinase 1; TRADD, TNF receptor associated death domain; cIAP, cellular inhibitor of apoptosis protein; TRAF, TNF receptor associated factor; FADD, Fas-associated death domain; CYLD, cylindromatosis; MLKL, mixed lineage kinase domain-like pseudokinase; CXCR4, C-X-C chemokine receptor type 4; uPA, urokinase-type plasminogen activator; TAK1, transforming growth factor-β (TGF-β)-activated kinase 1; MAPK, mitogen-activated protein kinase; CXCL, C-X-C chemokine ligand; CCL20, chemokine C-C motif ligand 20; CSF1, colony-stimulating factor 1; TRIM, tripartite motif-containing; GM-CSF, granulocyte-macrophage colony-stimulating factor; 5-FU, 5-fluorouracil; GSDM, gasdermin; DRD2, D2 dopamine receptor; cytC, cytochrome C; AMPK, AMP-activated protein kinase; SIRT1, silent mating type in-formation regulation 2 homolog-1; GPX4, glutathione peroxidase 4; RSL3, RAS-selective lethal 3; SLC7A11, solute carrier family 7, Member 11; GSH, glutathione; GSSG, oxidized glutathione; DMF, dimethylformamide; NKAP, NF-κB activating protein; LCN2, Lipocalin 2; LIFR, leukemia-inhibitory factor receptor; SHP1, Src homology domain 2-containing protein tyrosine phosphatase 1; DFX, deferasirox; HMGB1, high mobility group box 1.

**Figure 3 pharmaceuticals-16-00783-f003:**
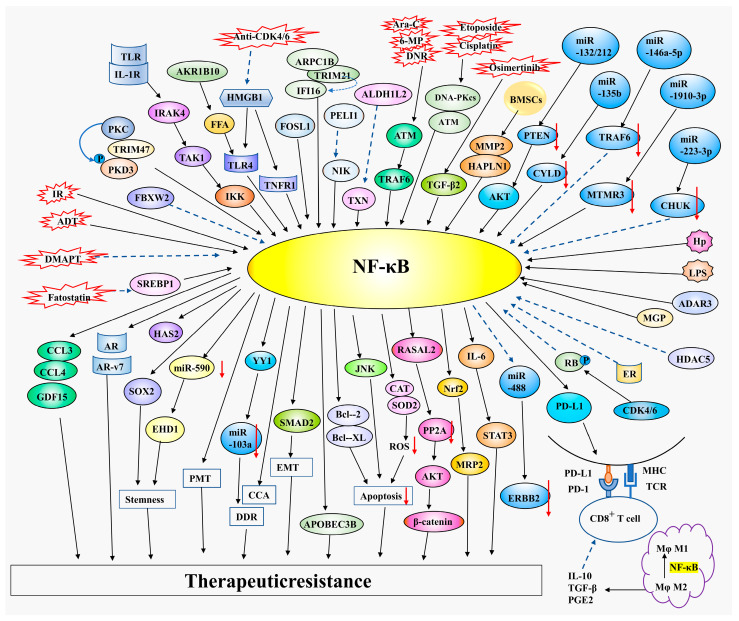
NF-κB and therapeutic resistance. Black solid arrows represent facilitation and blue dashed arrows represent inhibition. Hp, Helicobacter pylori; RASAL2, rat sarcoma (RAS) protein activator-like 2; PP2A, protein phosphatase 2A; NRF2, nuclear factor-erythroid 2 p45 related factor 2; MRP2, multidrug resistance-associated protein 2; EGFR-TKIs, epidermal growth factor receptor-tyrosine-kinase inhibitors; EHD1, C-terminal Eps15 homology domain-containing 1; ERBB2, receptor tyrosine-protein kinase 2; MTMR3, myotubularin-related protein 3; PTEN, phosphatase and tensin homolog deleted on chromosome 10; JNK, Jun N-terminal kinase; ATM, ataxia-telangiectasia mutated gene; DNA-PKcs, DNA-PK catalytic subunit; APOBEC3B, apolipoprotein B mRNA editing catalytic polypeptide 3B; FBXW2, F-box and WD-repeat-containing protein 2; SOX2, SRY-related high-mobility-group (HMG)-box protein 2; STAT3, signal transducer and activator of transcription 3; ADAR3, the third member of adenosine deaminase that acts on RNA; HAPLN1, hyaluronan and proteoglycan link protein 1; MMP2, matrix metalloproteinase 2; BMSCs, bone marrow stromal cells; Ara-C, cytarabine; DNR, daunorubicin; 6-MP, 6-mercaptopurine; GDF-15, growth differentiation factor 15; CHUK, conserved helix-loop-helix ubiquitous kinase; IR, ionizing radiation; AKR1B10, aldo-keto reductase B10; FFA, free fatty acid; CCA, cell cycle arrest; DDR, DNA damage repair; PELI1, pellino E3 ubiquitin protein ligase 1; PMT, proneural (PN) to mesenchy-mal (MES) transition; FOSL1, FOS-like antigen 1; ARPC1B, actin-related protein complex 1B; IFI16, γ-interferon inducible protein 16; TXN, thioredoxin; ALDH1L2, aldehyde dehydrogenase 1 family member L2; CAT, catalase; ER, estrogen receptor; TAM, tamoxifen; PKC-ε, protein kinase C-ε; PKD3, protein kinase D3; AR, androgen receptor; ADT, androgen-deprivation therapy; AR-V7, AR variant 7; DMAPT, dimethylaminoparthenolide; SREBP, sterol regulatory element-binding protein; PD-1, programmed death-1; PD-L1, programmed death- ligand 1; RB, retinoblastoma; CDK4/6, cyclin-dependent kinases 4 and 6; HDAC5, the histone deacetylase member 5; PGE2, prostaglandin E2; IRAK4, L-1R-associated kinase 4; HAS2, hyaluronan synthase 2; MGP, matrix Gla protein.

**Table 1 pharmaceuticals-16-00783-t001:** Regulation mechanisms and specific effects of bioactive molecules or exogenous compounds that promote or inhibit NF-κB in various tumor types.

Agents	The Function of NF-κB (PRO, Promotion; INH, Inhibition)	Regulation Mechanism	Specific Effects on Various Tumor Types	Reference
Polyphyllin VI	PRO	activates the NF-κB/NLRP3/caspase 1/GSDMD pathway, and promotes pyroptosis	Anti-NSCLC	[68]
Metformin	PRO	activates the AMPK/SIRT1/NF-κB/Bax-cytC/caspase 3/GSDME pathway, and promotes pyroptosis	Anti-HCC, breast cancer, and CRC	[69]
Lupeol	PRO	activates the AMPK/NF-κB pathway, decreases GPX4, and triggers ferroptosis	Anti-NPC	[78]
RSL3	PRO	activates NF-κB, decreases GPX4, and induces ferroptosis	Anti-GBM	[79]
H. pylori	PRO	activates the NF-κB/RASAL2/AKT/β-catenin pathway	Promotes gastric tumorigenesis, and chemoresistance of platinum and fluorouracil	[82]
Osimertinib	PRO	activates the TGFβ2/NF-κB/SMAD2/EMT pathway	Promotes osimertinib resistance in NSCLC	[85]
Etoposide and Cisplatin	PRO	activates the DNA-PKcs/ATM/NF-κB/APOBEC3B pathway	Promotes breast cancer progression and therapy resistance	[92]
LPS	PRO	activates the NF-κB/IL-6/STAT3 pathway	Promotes prostate cancer progression and docetaxel resistance	[93]
Ara-C, DNR, and 6-MP	PRO	activates the ATM/TRAF6/NF-κB/GDF15, CCL3, CCL4 pathway	Promotes drug resistance of ALL	[98]
Tanshinone II	INH	activates caspase 3/GSDMD axis and promotes pyroptosis	Anti-cervical cancer	[66]
Piperlongumine analogue L50377	INH	induces ROS-mediated NF-κB inhibition and promotes pyroptosis	Anti-NSCLC	[67]
DMF	INH	induces lipid peroxidation and ferroptosis	Anti-DLBCL	[72]
DFX	INH	inhibits ferroptosis, DFX + SOR promotes apoptosis	Anti-HCC	[76]
CDK4/6 kinase suppressors (Palbociclib, Ribociclib, and Abemaciclib)	INH	inhibits the HMGB1/TLR-4/NF-κB pathway	Reverses TAM resistance in breast cancer	[128]
DMAPT	INH	oral NF-κB inhibitor	Reverses resistance to ADT in prostate cancer	[140]
Fatostatin	INH	inhibits the SREBP1/NF-κB pathway	Reverses progesterone resistance in endometrial cancer	[141]

**Table 2 pharmaceuticals-16-00783-t002:** Targeting NF-κB nano delivery systems in various tumor types.

Agents or Nucleic Acid	NPs Encapsulation	Tumor Types	Reference
Curcumin + selenium + DOX	PSHA	CRC	[154]
DOX + CXB	HA-GC	NSCLC	[155]
Thiotetrazole	PLGA-PEG-HA	pancreatic cancer	[156]
Chrysin	PCL-PEG	breast cancer	[158]
Astraxanthin	SLN	breast cancer	[157]
Disulfiram	PLGA	GBM	[159]
siNF-κB	peptide	ATLL	[160]
siNF-κB	polymeric micelle	melanoma	[161]

## Data Availability

Data sharing not applicable.

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
