# Peer review of "NF-κB in Cell Deaths, Therapeutic Resistance and Nanotherapy of Tumors: Recent Advances"

_pharmaceuticals, 2023, doi:10.3390/ph16060783_

Round 1
Reviewer 1 Report
This is a review article entitled “NF-κB in Cell Deaths and Therapeutic Resistance of Tumors: Recent Advances”. The authors have nicely presented the role of NF-κB transcription factor in cancer resistance. They have described how NF-κB is major player in cancer resistance against different treatment modalities, such as chemotherapy, radiotherapy, endocrine therapy, immunotherapy etc.
However, I have 2 comments which authors should address –
1. The interplay between inflammation, NF-κB and cancer was not clearly described in the manuscript. Author should discuss it in detail.
2. Author should present a future outlook in the manuscript. How we can overcome the cancer therapeutic resistance by targeting NF-κB.
Author Response
Q1. The interplay between inflammation, NF-κB and cancer was not clearly described in the manuscript. Author should discuss it in detail.
Response: Thanks for your kind comments. We sincerely apologize for our negligence. As you mentioned, NF-κB is indeed the link between inflammation and cancer. NF-κB is involved in the whole process of tumor evolution through facilitating inflammation in carcinogenesis and regulating the inflammatory state of the tumor microenvironment in tumor progression. In recent decades, numerous high-quality literatures have already summarized the molecular mechanism and function of NF-κB in regulating inflammation during tumorigenesis and progression. And due to the limited space, we focused on the regulation of NF-κB in cell death and treatment resistance in various tumors. Please browse the revised manuscript for details. (Lines 488-494, Page 9)
Q2. Author should present a future outlook in the manuscript. How we can overcome the cancer therapeutic resistance by targeting NF-κB.
Response: Thanks for your excellent advice. According to your suggestions, we have added the future outlook behind the conclusion section (Lines 498-504, Page 9), please check the content in the revised manuscript.
Reviewer 2 Report
Xuesong Wu and colleagues addressed several aspects of NF-kB function in cancer cell death and therapy resistance in their review.
The article is well structured and enriched by images that help the reader understand the treated themes.
However, there are some aspects that should be improved.
Major concerns:
The epigenetic regulation mediated by miRNAs represents an intriguing field of study, especially for the formulation of novel therapeutic strategies. This theme is poorly addressed, and I think that it should be improved.
Accumulating evidence highlights the importance of the microbiota in the onset and progression of diverse human diseases, including cancer. The authors barely mention this interesting topic. I suggest improving this theme by connecting it to the part about Helicobacter pylori.
Authors describe NF-kB role in necroptosis, saying that "Necroptosis is similar to necrosis…", but what about necrosis? These additional details could be useful to further improve the contents of the review.
Minor concerns:
Correct the text for typos.
Author Response
Q1. The epigenetic regulation mediated by miRNAs represents an intriguing field of study, especially for the formulation of novel therapeutic strategies. This theme is poorly addressed, and I think that it should be improved.
Response: Thanks for your nice reminding. The relationship between NF-κB and miRNA is complicated. MiRNA interacts with NF-κB and eventually exerts the function of tumor promotion or prevention. According to your suggestion, we have supplemented the relevant information of miRNA in the section of tumor chemotherapy resistance (Lines 247-255, 258-262, 293-296, Page 5).
Q2. Accumulating evidence highlights the importance of the microbiota in the onset and progression of diverse human diseases, including cancer. The authors barely mention this interesting topic. I suggest improving this theme by connecting it to the part about Helicobacter pylori.
Response: Thanks for your excellent advice. In the revised version, we introduce the involvement of microbiota in tumor progression and link it with the Helicobacter pylori section in the original manuscript (Line 226-231, page4).
Q3. Authors describe NF-κB role in necroptosis, saying that "Necroptosis is similar to necrosis…", but what about necrosis? These additional details could be useful to further improve the contents of the review.
Response: We apologize for our negligence and have corrected the corresponding content (Lines88-89, Page2).
Reviewer 3 Report
The authors aim to present a review article entitled “NF-κB in Cell Deaths and Therapeutic Resistance of Tumors: Recent Advances”. This paper described and discussed the recent research on the regulation of NF-κB in cancer cell deaths and therapy resistance. The manuscript is nicely composed, and the figures are also presented in an appropriate way having a lot of new information. Before accepting the manuscript, authors are advised to address a few points:
1. The authors may briefly describe how they went about data collection prior to writing the review. This would add more value to the work.
2. Authors should mention and list in tabular form about few bioactive molecules or exogenous compounds facilitate/hamper NF-κB signaling for each function in the revised version.
3. During the review of the manuscript, it has been noticed that there was no description of the nano delivery systems of bioactive/exogenous compounds affecting NF-κB signaling. Authors should collect some information related to this and include in the revised manuscript.
4. Authors should check the legends of figures. Abbreviations should be common and same in the manuscript and legends.
5. Formatting, grammatical errors should be verified throughout the manuscript by the authors.
There are some formatting and grammatical errors. Authors should recheck and correct it throughout the manuscript.
Author Response
Q1. The authors may briefly describe how they went about data collection prior to writing the review. This would add more value to the work.
Response: Thanks for your valuable advice. The data collection process is added in the introduction (Lines 38-41, Page1).
Q2. Authors should mention and list in tabular form about few bioactive molecules or exogenous compounds facilitate/hamper NF-κB signaling for each function in the revised version.
Response: Thank you very much for your nice comments. We have supplemented this part to Table 1. Regulation mechanisms and specific effects of bioactive molecules or exogenous compounds that promote or inhibit NF-κB in various tumor types (Pages 12-13).
Q3. During the review of the manuscript, it has been noticed that there was no description of the nano delivery systems of bioactive/exogenous compounds affecting NF-κB signaling. Authors should collect some information related to this and include in the revised manuscript.
Response: We are grateful for the suggestion and we have added a detailed interpretation regarding the nano delivery systems of bioactive/exogenous compounds affecting NF-κB signaling. The details are added in the revised manuscript (Lines 18-19, 44-50, Pages1; Lines 443-486, Pages8-9; Lines 501-504, Pages9).
Q4. Authors should check the legends of figures. Abbreviations should be common and same in the manuscript and legends.
Response: Thank you for your nice advice. We have made corresponding revisions according to your suggestion.
Q5. Formatting, grammatical errors should be verified throughout the manuscript by the authors. Comments on the Quality of English Language There are some formatting and grammatical errors. Authors should recheck and correct it throughout the manuscript.
Response: We are very sorry for our poor writing. We have carefully scrutinized the entire manuscript and will polish our writing through professional English editing services.
Round 2
Reviewer 2 Report
The authors replied to all the questions posed by the reviewer.